Influence of temperature on developmental and biochemical traits of the red squat lobster Grimothea monodon (H. Milne Edwards, 1837) during early ontogeny

Quispe-Machaca Marco 1
Olavarría Luis 2
Torres Gabriela 3
http://orcid.org/0000-0001-7706-5126 Urzúa Ángel 4 5 aurzua@ucsc.cl
1 Programa de Doctorado en Ciencias mención Biodiversidad y Biorecursos, Facultad de Ciencias, Universidad Católica de la Santísima Concepción , Concepción, Biobío , Chile
2 Instituto de Fomento Pesquero (IFOP) , Talcahuano, Biobío , Chile
3 Alfred-Wegener-Institut für Polar und Meeresforschung , Helgoland, Schleswig-Holstein , Germany
4 Departamento de Ecología, Facultad de Ciencias, Universidad Católica de la Santísima Concepción , Concepción, Biobío , Chile
5 Centro de Investigación en Biodiversidad y Ambientes Sustentables (CIBAS), Universidad Católica de la Santísima Concepción , Concepción, Biobío , Chile
Moreira Daniel
Electronic publication date: 2025 Oct 22
Publication date: 2025
Volume: 13
Electronic Location ID: e20278
Received 2024 Dec 12; Accepted 2025 Oct 1
Copyright: © 2025 Quispe-Machaca et al.
Copyright year: 2025
Copyright holder: Quispe-Machaca et al.
License: This is an open access article distributed under the terms of the Creative Commons Attribution License, which permits unrestricted use, distribution, reproduction and adaptation in any medium and for any purpose provided that it is properly attributed. For attribution, the original author(s), title, publication source (PeerJ) and either DOI or URL of the article must be cited.
License URL: https://creativecommons.org/licenses/by/4.0/

Keywords: Humboldt current ecosystem, Elemental composition, Red squat lobster, Early ontogeny, Bioenergetic condition, Environmental temperature, Larval stage, Crustaceans culture

Funding: Agencia Nacional de Investigación y Desarrollo (ANID)-Subdirección de Capital Humano/Beca Doctorado Nacional/2023/Folio 21230722 BMBF 180034 PROYECTO USC-20102 INCA 210005 Fondo de Mantención de Equipamiento para la Investigación DII-UCSC 2-2023 This work was supported by Agencia Nacional de Investigación y Desarrollo (ANID)-Subdirección de Capital Humano/Beca Doctorado Nacional/2023/Folio (No. 21230722), Grant BMBF (No. 180034), PROYECTO: “Internacionalización Transversal en la UCSC: enfrentando los nuevos desafíos” (No. USC-20102), Ciencia Abierta en la UCSC: Grant INCA (No. 210005) and Fondo de Mantención de Equipamiento para la Investigación DII-UCSC: (Grant 2-2023). The funders had no role in study design, data collection and analysis, decision to publish, or preparation of the manuscript.

==============================
Temperature is one of the most important environmental factors that influence on the successful development and survival of decapod larvae. Our model species, the red squat lobster Grimothea monodon, has a wide biogeographic distribution in the Humboldt Current Ecosystem (HCE) and support important fishing activities. Recently, it has been described that juvenile and adult individuals of G. monodon (i.e., benthic phase of their ontogeny), present intraspecific variations in size, lifestyle, and nutritional condition, which could be modulated by the environmental conditions like temperature associated with depth. However, it is still unknown whether these intraspecific variations also occur during early ontogeny (i.e., planktonic larval phase). To investigate, we evaluated the effect of contrasting temperatures (i.e., cold: 12 °C vs. warm: 20 °C) on the developmental and biochemical parameters of larvae of the red squat lobster G. monodon under laboratory conditions. Our results show that differences were observed only in the development time and larval size of the larvae developed at the two experimental culture temperatures. No significant variations were recorded in mortality during the larval phase (i.e., from zoea I to megalopa), nor were significant variations detected in the biomass (dry weight) or the biochemical-elemental constituents (carbon, hydrogen, nitrogen) of an advanced larval stage (zoea V) at the two evaluated temperatures. Our findings suggest that during early ontogeny G. monodon presents intraspecific variability in its developmental traits along with a high physiological-energetic plasticity that allows it to survive and successfully cope with the temporal and spatial variations in seawater temperature that frequently occur in the HCE.

Introduction

Variations in temperature have a major influence on the development, growth and survival time of the early ontogenetic stages of crustaceans (Pörtner & Farrell, 2008) because this phase of early ontogeny, compared to the adult phase, is vulnerable to adverse periods of high and low ambient temperatures (Gutiérrez-Estrada & Pulido-Calvo, 2023). For example, warm temperatures can accelerate larval growth rates (Bermudes & Ritar, 2004) and trigger the appearance of deformities in the body structure of larvae as observed in decapod species from temperate and cold environments such as Romaleon setosum and Chaceon quinquedens (Zeng et al., 2020; Pérez-Pérez et al., 2023). In turn, cold temperatures can slow larval growth rates, prolonging the larval development time in the plankton, and consequently increasing their risk of predation and/or mortality before completing the larval cycle and reaching the benthic juvenile phase (Green & Fisher, 2004).

Decapod crustaceans inhabit a wide variety of marine environments, including habitats with extreme environmental conditions such as deep-sea areas, which are characterized by abrupt changes in temperature, light penetration, food availability, and dissolved oxygen levels (Costello, Cheung & De Hauwere, 2010; Zeng et al., 2020). These factors collectively influence the distribution, diversity, behavior, and physiology of marine organisms (Costello, Cheung & De Hauwere, 2010; Freitas et al., 2021). In this context, decapods with a biphasic life cycle (i: free-living pelagic larval phase, ii: benthic juvenile-adult stage) have been observed coping with variations in key environmental factors (temperature, food availability, oxygen, salinity) on the seafloor or demersal areas by changing their developmental and biochemical traits during their successive ontogenetic stages (Almeida, Flores & Queiroga, 2008; Raventos et al., 2021). In particular, in decapods, it has been described that the maternal provision of energy to eggs, together with the prevailing environmental conditions during the embryogenesis phase, greatly influence the subsequent larval development (Weiss et al., 2009; Baldanzi et al., 2018). In this context, temperature is one of the most important environmental factors due to its high influence on the successful development and survival of larvae (Bermudes & Ritar, 2004). This physical parameter reflects kinetic energy and is considered one of the main modulators of environmental variability and climate change in marine ecosystems (Storch et al., 2009), especially influencing the distribution, abundance and survival of ectothermic marine organisms (Bhaud et al., 1995; Freitas et al., 2021).

Crustacean larvae are particularly sensitive to changes in environmental conditions, especially water temperature, which has a direct effect on their survival and growth (Ryer et al., 2016; Marochi, Duarte & Costa, 2024). The larval phase of decapod crustaceans consists of a series of ontogenetic stages that are completed by successive molts (Anger, 1996). This growth cycle is comprised by two components: (i) the duration of the intermolt period, defined as the interval between successive molts, and (ii) the molt increment, representing the increase in size and weight attained at each stage (Stoner, Ottmar & Copeman, 2010; Yuan et al., 2017). Thus, each species presents a number and/or frequency of molts determined according to its family (e.g., Jasus edwardsii: four larval stages (Bermudes & Ritar, 2004); Taliepus dentatus: two larval stages (Fagetti & Campodonico, 1971a)); Grimothea monodon: five larval stages (Fagetti & Campodonico, 1971b); Munida subrugosa: five larval stages (Roberts, 1973; Wehrtmann & Báez, 1997). The duration of these larval stages can change depending on the temperature (Anger, 1996; Hartnoll, 2001). From a physiological perspective, it has been well documented that high temperatures can generate a significant energy imbalance, greatly influencing the larval growth potential of decapods (Bermudes & Ritar, 2004). High temperatures increased mortality rates during the early stages (Koumoundouros et al., 2001). The biochemical and/or elemental composition (carbon, hydrogen, nitrogen) and bioenergetic ratios (C/N, C/H) (Diez et al., 2012) of larvae can also be affected by increased temperatures, as they can indicate their nutritional condition (Anger, 2001; Quispe-Machaca et al., 2024).

Our model species, the red squat lobster Grimothea monodon, has a wide biogeographic distribution from the Lobo de Afuera Islands in Perú to the Chiloé Islands in Chile. However, its range has been increasing, and its presence has been reported as far away as the coasts of the United States (DecaNet, 2024). This species has been of great commercial importance in demersal fisheries since 1950 (Guzmán et al., 2020), along with other squat lobster species such as G. johni. In Chile, there are fishing units dedicated to its extraction (Northern Fishing Unit: NFU; Southern Fishing Unit: SFU) (Yannicelli et al., 2012; Guzmán-Rivas, Quispe & Urzúa, 2022), with the largest extraction area in the SFU, specifically off the coast of Concepción in the Biobío Region (Palma & Arana, 1997; IFOP, 2022). The reproductive period of this species extends from autumn to spring with numerous broods from which numerous planktotrophic zoea larvae hatch (Rivera & Santander, 2005; Yannicelli et al., 2012; Barros, Alarcón & Arancibia, 2023). These larvae remain at depths between 50 and 100 m (Yannicelli, 2005) and face temporal and spatial variations in temperature that can influence their survival, and consequently, as a cascading effect, impact the recruitment rates of juveniles (Yannicelli et al., 2012). In this context, some studies have indicated that environmental conditions experienced during the early ontogeny of G. monodon may influence not only the nutritional status of individuals (Seguel et al., 2019; Guzmán-Rivas, Quispe & Urzúa, 2022), but also their lifespan. Individuals with better nutritional conditions and longer lifespans are found in the cold, rather than warm zones of the Humboldt Current Ecosystem (HCE) (Yapur-Pancorvo et al., 2023).

During the early days of development, decapod larvae require food that provides a substantial energy supply, which is essential for tissue and organ formation during growth (Vargas-Ceballos et al., 2020); however, studies on the physiological capacity of G. monodon larvae have not yet been carried out (Yannicelli et al., 2012). The ontogeny of the red squat lobster studied in the laboratory has demonstrated five larval stages (zoea I-Zoea V) and a megalopa before reaching the juvenile stage and later adult stage (Fagetti & Campodonico, 1971b; Yannicelli, 2005). However, as in other galatheids, the larval development of G. monodon to the megalopa stage can be variable due to its plastic development (Haye et al., 2010; Yannicelli et al., 2012). Sea temperature can thus influence the development and growth period of larvae, considering that the first stages of development in crustaceans are considered the most vulnerable part of their life cycle (Anger, 2001; Weiss et al., 2009).

Recently, it has been described that juvenile and adult individuals of G. monodon, corresponding to the benthic phase of their ontogeny, present intraspecific variations in size and/or lifestyle (Guzmán-Rivas, Quispe & Urzúa, 2022; Yapur-Pancorvo et al., 2023), which could be modulated by the environmental conditions of their habitat such as temperature, planktonic food availability, and oxygen level (Haye et al., 2010; Guzmán-Rivas, Quispe & Urzúa, 2022; Quispe-Machaca et al., 2024). However, it is still unknown whether these intraspecific variations also occur during early ontogeny (larval phase). Given that the red squat lobster G. monodon is an ectothermic invertebrate with a complex life cycle and highly sensitive larval development, inhabiting the temperate–cold latitudes of the HCE, we hypothesized that exposure to warmer temperatures than those of its natural habitat would increase larval energy expenditure, thereby affecting survival, development time, size, biomass, and elemental composition during early ontogeny. Therefore, the objective of the present work was to evaluate the effect of contrasting temperatures (cold vs. warm) on the growth, survival and biochemical parameters (C, H, N) of larvae of the red squat lobster G. monodon from SFU under laboratory conditions.

Materials and Methods

Collection site and maintenance of adult individuals

Adult individuals of G. monodon were collected in Faro Carranza (35°26′S 75°29′W), a marine area located on a wide continental shelf characterized by pronounced seasonal temperature variation and upwelling, which together generate high primary productivity (4–9 g C m−2d−1) (Daneri et al., 2000; Eissler et al., 2010). The collected squat lobsters were transported in 200 L seawater tanks to the Marine Biological Station Abate Juan Ignacio Molina (Fig. 1). In the laboratory, both males and females were placed together in 100 L tanks with continuous sea water flow. They were exposed to similar environmental conditions of temperature (12.81 °C), salinity (33–34 PSU) and photoperiod (12 h L:D); and they were fed ad libitum food with fresh mussel and fish pellets until the release of their ovarian load (female) and seminal ducts (male). Then, once the individuals had mated under laboratory conditions (starting a new reproductive cycle in captivity), the egg-bearing females (n = 3; carapace length: 50.3–55.5 mm) were placed in individual 40 L tanks and maintained under the same conditions mentioned above until larval hatching (larval stage: zoea I).

Figure 1 Sampling area of G. monodon in the HCE (~35 °S: Faro Carranza, Chile).

Larval culture

The larvae zoea-I were placed were placed in 1-L culture chambers containing filtered seawater that had been sterilized with ultraviolet light to eliminate protozoans that could compromise larval survival (Brown & Russo, 1979; Ford, Xu & Debrosse, 2001). The larvae were subjected to two temperature treatments: i. cold temperature (12 °C) and ii. warm temperature (20 °C). The experimental temperatures were established based on a projected increase of approximately 3 °C in the average cold-winter (9 °C) and warm-summer (17 °C) conditions under climate warming and marine heatwave scenarios for the study area (IPCC, 2022; Rahmstorf, 2024). A total of 11 culture chambers, each with a volume of 500 mL, were used in each treatment (12 °C vs. 20 °C). A group of 50 larvae (total N = 1,100 larvae) were placed in each culture chamber. During the development of experiments, the seawater was changed daily with new filtered and sterilized water to avoid the presence of microorganisms such as microalgae and protozoans that affect the survival of larvae. Finally, the larvae were fed daily with nauplius larvae of brine shrimp as live food (Artemia sp. Yannicelli et al., 2013).

Development, survival and larval size

The time and stage of development of the successive larval instars were evaluated by observation under a stereo microscope (Motic BA-310 model) and by determining molts (presence of exuviae) in the culture chambers, following the morphological description of the G. monodon larvae proposed by Fagetti & Campodonico (1971b). Larvae that reached a new instar were transferred to new culture chambers. In turn, to quantify survival (measured as a function of the % of mortality), every day the culture chambers were observed and the presence of dead individuals (larvae on the bottom with no signs of life) were recorded. Finally, to determine larval size (body length from posterior edge of eyestalk to posterior mid dorsal edge of carapace) (Rasmussen & Aschan, 2011) of the successive larval instars cultured at temperatures of 12 °C and 20 °C, a microscopy with a digital camera graduated with a ruler of 1mm and photographic records were used.

Biomass and elemental composition of larvae

To analyze the biomass (dry weight: DW) and elemental composition (carbon, hydrogen, and nitrogen contents: C, H, N) the standard technique of Anger & Harms (1990) was performed. The quantifications of DW and C, H, and N were performed only in samples of the larval stages zoea I and zoea V due to the greater availability of larval specimens (minimum sample size for analysis) necessary for these analyses. In turn, the larvae selected for analyses were obtained from larval stage zoea I one day after being subjected to the two temperature treatments after hatching, while larvae from stage zoea V were obtained immediately upon reaching this stage. For laboratory analyses, an appropriate number of larvae was pooled to obtain the sample mass required for DW and CHN determinations (i.e., 0.44–0.95 mg dry weight; 60–70 zoea-I larvae; 3–5 zoea-V larvae) (Yannicelli et al., 2013; Anger, Harzsch & Thiel, 2020). For larval biomass (DW) determinations, samples were placed in pre-weighed tin cartridges, lyophilized (model FDU-7012, Operon) and weighed on a microbalance (Perkin Elmer Instruments AD 6000). To determine the larval elemental composition (CHN), a Perkin Elmer Instruments elemental analyzer (series II CHNS/O Analyzer 2400) was used, which has an auto-sampler of up to 60 samples; acetanilide was used as a standard. The samples were incinerated at a temperature of 975 °C for approximately 5 min using helium gas as a carrier gas and oxygen as combustion (Viña-Trillos, Brante & Urzúa, 2023).

Statistical analysis

To evaluate the effect of the culture temperature (12 °C vs. 20 °C) on the development time and size of successive larval stages of G. monodon, a generalized linear model (GLM) with Gaussian distribution and an analysis of variance (ANOVA) were applied following standard methods (Zuur, Ieno & Elphick, 2010). Prior to the ANOVA analysis, the assumptions of normality, homoscedasticity and effect size were checked according to Zuur, Ieno & Elphick (2010). In turn, the effect of the culture temperature on larval survival (measured as a function of the % of mortality) was analyzed using the nonparametric Kaplan Meier method (Rodrigues, De Almeida & Bertini, 2018). Finally, variations in biomass (DW), elemental biochemical constituents (C, H, N) and bioenergetic ratios (C/N, C/H) of an initial larval stage (zoea I) vs. advanced (zoea V) were analyzed by T-test.

Results

Development, survival, and larval size

A total of 1,100 zoea I larvae were obtained, and all individuals completed the larval development cycle in the two temperature treatments. The maximum larval development time (zoea I-megalopa) was 110 and 91 days at a temperature of 12 °C and 20 °C, respectively (Table 1). During the experiment, possible new larval stages were observed (intra-individual variability) as a plastic response to the 12 °C temperature treatment. In the zoea-III stage, no changes were observed in the number of spines during molting; only an increase in the size of the internal uropods was recorded. Similarly, in the zoea-IV A stage, individuals that initially presented three setae molted with the addition of one seta but did not reach the zoea-IV B stage after molting (Fig. 2).

Table 1 Development time (days) of successive larval stages of G. monodon cultured at two temperatures (cold: 12 °C vs. warm: 20 °C).

Larval stage	12 °C	20 °C	
Min	Max	X¯ ± SD	Min	Max	X¯ ± SD	
Zoea I	3	22	8.55 ± 3.30	4	17	6.00 ± 2.49	
Zoea II	15	35	17.60 ± 2.72	9	26	10.60 ± 1.17	
Zoea III	24	48	27.00 ± 3.12	12	36	14.80 ± 2.86	
Zoea IV-A	32	67	40.38 ± 9.78	17	52	20.56 ± 4.42	
Zoea IV-B	40	89	46.38 ± 8.78	21	64	26.56 ± 6.54	
Zoea IV-C	49	104	56.75 ± 12.35	30	75	33.56 ± 6.67	
Zoea IV-D	58	104	62.86 ± 7.93	34	77	43.00 ± 13.78	
Zoea IV-E	68	104	74.00 ± 8.58	40	91	48.75 ± 13.99	
Zoea V	78	111	90.86 ± 11.29	47	92	59.63 ± 13.77	
Zoea VI	107	110	108.50 ± 2.12	54	91	62.50 ± 10.99	
Megalopa	110	0	110.00	54	82	73.50 ± 14.68	
Note:

Min, Minimum; Max, Maximum; X¯, Mean; SD, Standard Deviation.

Figure 2 Larval cycle (from zoea I to megalopa) of G. monodon cultured at two temperatures ((A) cold: 12 °C vs. (B) warm: 20 °C).

Symbol (?) indicates possible plastic response to cold temperature.

Larval survival (measured as the % of mortality) in both temperature treatments (12 °C vs. 20 °C) showed similar trends until day 12 (~45%). In turn, during larval development at 12 °C, the % of mortality increased from day 13 to day 33 and then remained stable until day 77. From this day on, a similar percentage of mortality was observed in both temperature treatments (i.e., mortality curves overlapped) (Fig. 3). Finally, when comparing the mortality curves of red squat lobster larvae subjected to temperatures of 12 °C and 20 °C using the nonparametric Kaplan-Meier test, these showed no significant differences during their development time (χ2: p = 0.071).

Figure 3 Mortality curves of red squat lobster larvae cultured at two temperatures (cold: 12 °C vs. warm: 20 °C).

The larval development time at the cold temperature of 12 °C was observed to be longer than that at the warm temperature (Fig. 4A). When comparing the effect of temperature on the larval size of successive instars, GLM results indicated that the interaction between larval stage and temperature was not significant (p = 0.051), with differences detected only among larval stages (p < 0.05) (Table 2; Table S1). However, when the effect of temperature on larval size was evaluated independently, the larvae cultured at the cold temperature of 12 °C were found to be significantly larger than those cultured at the warm temperature of 20 °C (Fig. 4B, Table 3). This variability in larval size showed statistical differences among the three larval stages, zoea IV-A (ANOVA: F1,18 = 16.767, p = 0.0007), zoea IV-B (ANOVA: F1,18 = 4.531, p = 0.0474) and zoea IV-E (ANOVA: F1,18 = 20.78, p = 0.0002).

Figure 4 (A) Development time and (B) size of successive larval stages of G. monodon cultured at two temperatures (cold: 12 °C vs. warm: 20 °C).

The asterisk in red (*) shows significant differences.

Table 2 Predictor variables included in the fitted Gaussian model.

	Df	Deviance residuals	Df residuals	Deviance	F	p	
NULL			190	27.13			
Temperature	1	0.01	189	27.12	1.89	0.17	
Larval stage	9	26.17	180	0.96	571.99	<2e−16	
Temperature: Larval stage	9	0.09	171	0.87	1.93	0.051	
Notes:

Df, Degree Freedom.

Significant effects (p < 0.05) are shown in bold.

Table 3 Size of successive larval stages of G. monodon cultured at two temperatures (cold: 12 °C vs. warm: 20 °C).

Stage larval	12 °C	20 °C	p-value	
X ± SD (ug)	X ± SD (ug)	
Zoea I	0.68 ± 0.05	0.65 ± 0.02	p = 0.26	
Zoea II	0.79 ± 0.04	0.75 ± 0.05	p = 0.08	
Zoea III	0.87 ± 0.08	0.82 ± 0.05	p = 0.12	
Zoea IV-A	1.02 ± 0.07	0.90 ± 0.05	p = 0.0007	
Zoea IV-B	1.20 ± 0.10	1.11 ± 0.09	p = 0.047	
Zoea IV-C	1.35 ± 0.09	1.27 ± 0.09	p = 0.07	
Zoea IV-D	1.44 ± 0.08	1.40 ± 0.09	p = 0.35	
Zoea IV-E	1.65 ± 0.04	1.50 ± 0.10	p = 0.0002	
Zoea V	1.74 ± 0.10	1.66 ± 0.09	p = 0.06	
Zoea VI	1.74 ± 0.08	1.79 ± 0.04	ND	
Note:

ND, No Data (not enough replicates of zoea VI at 12 °C).

Biomass and elemental composition of larvae

Comparisons between the analyzed larval stages (zoea I vs. zoea V) for each temperature (12 °C, 20 °C), revealed a significant increase in the absolute values (ug/ind) of larval biomass (DW) and its elemental constituents (C, H, N) (Fig. 5A, Table 4). In turn, in the relative values (%) of the larval elemental composition for each temperature, only the % of C showed a significant decrease in zoea V, while the % of N and H remained similar between the analyzed larval stages. When comparing between temperatures (12 °C vs. 20 °C) and considering only the zoea V stage, a slight increase was reported in the absolute (ug/ind) and relative (%) values of biomass (DW) and elemental composition (CHN); however, these differences were not statistically significant (see Fig. 5B, Table 4).

Figure 5 Elemental composition (carbon, hydrogen, nitrogen; CHN) of zoea I and zoea V of G. monodon cultured at two temperatures (cold: 12 °C vs. warm: 20 °C).

(A) absolute (ug/ind) and (B) relative (%) values.

Table 4 Biomass (ug/ind), elemental composition (CHN) and bioenergetic ratios (C/N, C/H) of zoea I and zoea V of G. monodon cultured at two temperatures (cold: 12 °C vs. warm: 20 °C).

	12 °C	20 °C	
Elemental composition	Zoea I	Zoea V	p-value	Zoea I	Zoea V	p-value	p-value	
X̄ ± SD	X̄ ± SD	(between stage to 12 °C)	X̄ ± SD	X̄ ± SD	(between stage to 12 °C)	Zoea V (12 °C vs. 20 °C)	
Dry Weight (ug/ind)	7.74 ± 1.06	197.17 ± 33.52	p = 0.0006	7.30 ± 1.06	189.21 ± 35.33	p = 0.0019	p = 0.7757	
C (ug/ind)	2.13 ± 0.35	35.55 ± 9.68	p = 0.0029	2.12 ± 0.34	38.42 ± 2.95	p = 0.0001	p = 0.9796	
H (ug/ind)	0.25 ± 0.05	6.15 ± 1.76	p = 0.0044	0.24 ± 0.05	5.76 ± 0.73	p = 0.0006	p = 0.6759	
N (ug/ind)	0.32 ± 0.06	7.44 ± 2.64	p = 0.0094	0.34 ± 0.06	7.61 ± 0.58	p = 0.0001	p = 0.9235	
%C	27.55 ± 2.55	19.81 ± 1.81	p = 0.0104	29.07 ± 3.69	20.61 ± 2.20	p = 0.0122	p = 0.4599	
%H	3.18 ± 0.30	3.08 ± 0.42	p = 0.7556	3.26 ± 0.56	3.05 ± 0.19	p = 0.5030	p = 0.9207	
%N	4.09 ± 0.49	3.69 ± 0.79	p = 0.5013	4.69 ± 0.65	4.09 ± 0.50	p = 0.2200	p = 0.4466	
C/N	6.76 ± 0.24	5.35 ± 0.71	p = 0.0305	6.21 ± 0.13	5.05 ± 0.21	p = 0.0004	p = 0.4560	
C/H	8.69 ± 0.61	6.32 ± 0.30	p = 0.0037	8.96 ± 0.45	6.74 ± 0.37	p = 0.0008	p = 0.1679	
Note:

Absolute (ug/ind) and relative (%) values. P-value in function to T-test. C, Carbon; H, Hydrogen; N, Nitrogen; %C, percentage of carbon; %H, percentage of Hydrogen; %N, percentage of Nitrogen; X¯, Mean; SD, Standard Deviation.

At the bioenergetic ratio level, at both temperatures (12, 20 °C) the C/N and C/H values of the advanced larval stage (zoea V) presented significantly lower values compared to the initial larval stage (zoea I) (Fig. 6, Table 4). In turn, when comparing between temperatures (12 °C vs. 20 °C) and considering only the zoea V stage, the C/N ratio presented similar values between the experimental cultivation temperatures. For the C/H ratio, zoea V larvae cultured at 20 °C presented slightly higher values than those cultured at 12 °C. Consequently, there was no statistically significant effect of culture temperature on the bioenergetic ratios of zoea V (T-test: C/N: t (5:0.05) = −0.81, p = 0.456; T-test: C/H: t (5:0.05) = 1.61, p = 5.00) (Fig. 6, Table 4).

Figure 6 Bioenergetic ratios of zoea I and zoea V of G. monodon cultured at two temperatures (cold: 12 °C vs. warm: 20 °C).

(A) C/N, (B) C/H.

Discussion

Temperature promotes changes in the metabolic activity of marine invertebrates with high thermal sensitivity (Bennett et al., 2019; Czaja et al., 2023). This affects their reproductive and developmental parameters, including fertilization rates, and embryonic and larval development (Estrada-Godínez et al., 2015). For decapods, temperature changes can negatively affect their generation time, while accelerating physiological processes such as growth and reproduction and increasing intraspecific variability rates (Willig & Presley, 2018). Due to this, the environmental temperatures exert selective pressure on population phenotypes (Lardies, Arias & Bacigalupe, 2010; Barria et al., 2018). These responses are important for maintaining the fitness of the species in relation to the environment (Storch et al., 2009), as observed in our findings in G. monodon, which proved that depending on the water temperature (cold vs. warm) intraspecific variability during larval development occurred.

Since temperature influences the life history traits of decapod crustaceans, its effect may be noticeable during the early ontogeny of species that support important fishing activities (Fischer & Thatje, 2016), such as our model species G. monodon (Guzmán et al., 2020). In this context, in a previous study by Fagetti & Campodonico (1971b) on the larval development cycle of G. monodon, larvae were cultured at two temperatures (15 °C vs. 20 °C) and a total of five larval stages (zoea I–V), including some larval substages at the zoea IV level, were reported. These are similar to our findings, where G. monodon larvae were cultured at two temperatures (12 °C vs. 20 °C), and a total of 6 larval stages (zoea I-VI) were observed, along with some larval substages at the zoea IV level (Fig. 2). Therefore, these comparative findings from the larval cycle of G. monodon indicate that temperature may play a modulating role in the intraspecific variability in early ontogeny traits (i.e., total number of larval stages, morphological varieties in larval substages) of G. monodon.

It has been well documented those contrasting temperatures (cold vs. warm) can have a significant effect on the development time, growth and/or larval size of decapod crustaceans (Anger, 2001), consequently altering the periods and frequencies of molts and inter-molts during their larval cycle (Ren et al., 2021; Khalsa et al., 2023). In particular, in our findings on the larval development time of G. monodon, at the cold temperature (12 °C) larvae developed slower than at the warm temperature (20 °C) (Fig. 4A). From a physiological approach, the larvae in warmer environments could develop more rapidly, as reported for zoea I and II stages by Yannicelli et al. (2013), due to an increase in their metabolic rates and the activity of molting enzymes and hormones that occur at high and/or warm temperatures, subsequently impacting growth processes (Navarro-Ojeda, Cuesta & González-Ortegón, 2021). Finally, this greater metabolic and/or energetic demand at the warm temperature (20 °C) was reflected by a reduction in the size of successive stages during the larval cycle of G monodon, as reported in our study (Fig. 4B) and in other species of decapod crustaceans from cold-temperate environments (Baudet et al., 2022).

Temperature has also been shown to influence the survival and/or mortality of the initial stages of the life cycle of ectothermic invertebrates such as decapod crustaceans (Ren et al., 2021; Khalsa et al., 2023). In the case of G. monodon, Fagetti & Campodonico (1971b) reported higher mortality at warm (20°C) than cold temperatures (15 °C) during the larval culture. These findings are contrary to our results because, although different percentages of mortality were found during larval development depending on the culture temperature, these differences were not statistically significant (Fig. 3). This finding could indicate that G. monodon larvae present compensatory physiological mechanisms that allow them to cope with the energy costs linked to temperature changes (Ren et al., 2021). In this context, it is proposed that, depending on the culture temperature and the type of food offered, G. monodon larvae could differentially use their main biochemical and/or elemental constituents as bioenergetic fuel. This compensatory mechanism has been described in several species of decapods (Litopenaeus vannamei: He et al., 2018; Crangon crangon: Anger, 2001; Urzúa et al., 2012; Anger, Harzsch & Thiel, 2020), which, depending on the temperature, have presented a degree and sequence of use and/or assimilation of carbohydrates (inferred from C and H), lipids (based on C), and proteins (quantified by N) (Sterner & Elser, 2017).

The red squat lobster G. monodon is an important marine bioresource along the Chilean coast, characterized by the high reproductive potential of the female stock, with average densities reaching 74 million eggs per square kilometer (Barros, Alarcón & Arancibia, 2023; Yapur-Pancorvo et al., 2023). Our species of interest has an indirect life cycle. Females incubate their eggs in the abdomen where the embryos develop under the same environmental conditions as the mother’s habitat (temperature, salinity, dissolved oxygen) (Urzúa et al., 2018), and from which numerous free-living planktonic larvae hatch (Yannicelli et al., 2012). Therefore, the survival of the new larvae (zoea I) depends on the first energy reserves they obtain from the egg, which are rapidly catabolized in the absence of food (Guerao et al., 2012). In turn, the energy reserves of the successive later larval stages depend on the food consumed and deposited in their body biomass (Anger, 2001). Consequently, in our findings, for each temperature the variations in the percentages of C (as a proxy for lipids) between larval stages may indicate the storage and subsequent use of this biomolecule as the main source of energy to meet the energy demands that occur during the larval cycle (Viña-Trillos, Brante & Urzúa, 2023). On the contrary, for each temperature the % of N and H remained relatively stable between stages (without variation). This finding could indicate the preservation of these biomolecules, especially N, as a proxy for proteins destined to form body structures (musculature) necessary for the development and growth of the larvae (Salonen et al., 1976; Ikeda et al., 2011; Urzúa et al., 2012).

In turn, at the level of bioenergetic ratios, at both temperatures (12 °C, 20 °C) the C/N and C/H values found during the advanced larval stage (zoea V) were significantly lower compared to those at the initial larval stage (zoea I). This decrease in bioenergetic ratios, as larval ontogenetic development progressed, can be considered a sign of the use of these components as a metabolic substrate to support the greater demand or energy expenditure that increases as a function of growth during early ontogeny in decapods (Anger, 2001; Anger & Moreira, 2004).

Conclusions

In our experiment, which evaluated the effect of contrasting temperatures (cold: 12 °C; warm: 20 °C) on the developmental and biochemical parameters of red squat lobster (G. monodon) larvae, differences were observed only in development time and larval size. No significant variations were recorded in mortality during the larval phase (from zoea I to megalopa), nor were significant variations detected in the biomass (DW) or the biochemical-elemental constituents (CHN) of an advanced larval stage (zoea V) at the two evaluated temperatures. These results suggest that during early ontogeny G. monodon presents intraspecific variability in its developmental traits along with a high physiological-energetic plasticity that allows it to survive and successfully cope with the temperature variations that frequently occur in planktonic marine environments of temperate-cold latitudes such as the HCE. Overall, our findings allow us to update our knowledge on the influence of temperature on the development, growth, survival and larval bioenergetic patterns of G. monodon in the HCE, an important commercial resource as it is the main resource of the industrial crustacean fishery. This information is key to improving our understanding of the intricate relationships that exist between the larval phase and juvenile recruitment, subsequently determining the adult stock exploited by industrial fisheries.

Supplemental Information

Supplemental Information 1 Raw data of Development time, Mortality percentage, Carapace length of larvae (LCap_larva) and elemental composition (CHN).

Supplemental Information 2 Summary of the generalized linear model (GLM) fitted to larval stage (I–VI) and temperature (12 °C vs. 20 °C). Significant effects (p < 0.05) are shown in bold.

Special thanks to Christine Harrower for correcting the English and improving this manuscript, to Wladimir Escalante for his guidance and advice on culture of crustacean larvae and Esthefany Reyes for her helpful in the chemical analyses. We sincerely thank reviewers for their constructive criticism and important suggestions.

Additional Information and Declarations

Competing Interests

Luis Olavarría is employed by Instituto de Fomento Pesquero (IFOP).

Author Contributions

Marco Quispe-Machaca conceived and designed the experiments, performed the experiments, analyzed the data, prepared figures and/or tables, authored or reviewed drafts of the article, and approved the final draft.

Luis Olavarría performed the experiments, prepared figures and/or tables, and approved the final draft.

Gabriela Torres performed the experiments, analyzed the data, authored or reviewed drafts of the article, and approved the final draft.

Ángel Urzúa conceived and designed the experiments, performed the experiments, analyzed the data, prepared figures and/or tables, authored or reviewed drafts of the article, and approved the final draft.

Data Availability

The following information was supplied regarding data availability:

The raw data are available in the Supplemental File.

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
