# Peer review of "Influence of temperature on developmental and biochemical traits of the red squat lobster Grimothea monodon (H. Milne Edwards, 1837) during early ontogeny"

_PeerJ, doi:10.7717/peerj.20278_

## Round 0.1 · original submission · Major Revisions

We invite you to submit a revised version that addresses the reviewers’ comments or provides rebuttals where appropriate.

Key points raised by the reviewers include: (i) expanding the temperature range tested (e.g., 26–30°C) and increasing the number of specimens to strengthen statistical robustness; (ii) exploring additional statistical approaches, such as regression analysis and multivariate methods (e.g., DistLM, RDA, GLM), to provide deeper insights into the data; (iii) revising the introduction to emphasize temperature earlier and ensuring smoother transitions between ideas; and (iv) discussing certain results in greater detail and ensuring they are appropriately placed within the Discussion section.

In addition, the reviewers suggested minor adjustments to figures and language refinement to improve readability. We encourage you to address these where feasible or acknowledge them as limitations.

Reviewer 1 ·

Basic reporting

General Comments
I have reviewed the manuscript “Influence of temperature on developmental and biochemical traits of early ontogeny of the red squat lobster Grimothea monodon (H. Milne Edwards, 1837)” as requested.
The key finding was that larval squat lobsters experience different rates of ontogenetic change depending on temperature, which is associated with depth. The results imply a degree of plasticity for developing squat lobster larvae. These findings will help to determine life history trajectories and biological parameters which underpin the squat lobster fishery.
The manuscript fills a novel data gap in the literature. The study was well-conceived and experimental design was appropriate. The figures are generally well presented.
I have minor comments about word choice, grammar and organization of the paper and minor suggestions about the figures. I suggest bringing temperature up sooner in the introduction, since it is the focal point of the paper. There also needs to be clearer linking sentences between ideas, since in many places ideas seem to pop up unexpectedly. Some aspects of the results appear to belong in the discussion. I provide some line by line comments below to indicate why I consider the paper is not yet suitable for publication, though these aspects should be simple enough to address. I would happily review a revised version.

Specific Comments
Title
I suggest you simplify the title to;
“Influence of temperature on developmental and biochemical traits of the red squat lobster Grimothea monodon (H. Milne Edwards, 1837) during early ontogeny”

Abstract
Ln 23. Decapod should not be pluralized
Ln 29. Why not just say “Which could be modulated by environmental conditions like temperature” to use simpler language and bring it back to your key variable of interest which is temperature associated with depth
Ln 30-31. Why not just say “To investigate, we evaluated the effects of contrasting temperatures…” to use simpler and more straightforward language
Ln 33. Poor wording. Why not just say “Our results show…”
Ln 35. Remove “in turn”

Introduction
Ln 45. What is important about deep sea areas? You could link these ideas to show the reader why this matters.
Ln 48. Remove “where they live”
Ln. 53. As temperature is the key aspect, this could come up sooner
Ln 61. Why not just say vulnerable
Ln 62-69. Seems like this is your key point
Ln 70. Why are you now telling the reader about molts? This needs a linking sentence to temperature much sooner than Ln 76
Ln 79. Why not just say increased. Simpler language and less wordy terms would be better here and throughout the manuscript
Ln 80. Capitalize carbon, hydrogen, nitrogen to show the reader that this is what you refer to as C/H/N etc. from here on
Ln 80. Add a space to the start of the sentence
Ln 82. Reduce sentence length here and throughout; for instance, there is no need to say “considered by researchers”. Why not just say “as they can indicate nutritional condition”?
Correcting these kinds of wordy sentences here and throughout would make the paper more succinct and improve readability.
Ln 103. This short paragraph needs a better topic sentence
Ln 118. Rather than telling the reader the object of your work, I would prefer if you stated your hypotheses upfront and more clearly in this section. I see that you do state a hypothesis at Ln 122, but in my opinion this should be the start of your aims section. You should also briefly state what you did and your study region
Ln 121. “Provided that?” is confusing language, it implies there was some doubt that G. monodon is ‘an ectothermic invertebrate that inhabits temperate-cold latitudes’. I do not think this is what you meant. How about “Since” instead ?

Materials and Methods
Ln 127. I suggest you add a section describing the collection site
Ln 135. Should be an em-dash here and throughout when you are giving a range of numbers
Ln 138. were previously placed ?
Ln 139. Remove ‘in the experimental design’ . If you must have subheadings, consider making experimental design a subheading for this section.

Results
Ln 188. Change to “and all individuals completed the larval development cycle” or similar
Ln 191. Why not just say “during the experiment”
Ln 193. Why not just say “No change was observed in the zoea III stage” or similar
Ln 195. This seems more like a discussion point than results
Ln 197. Why not just say “X individuals were unable to complete the larval cycle” or similar. Don’t discuss your results in this section, just present them
Ln 200. Change to “until day 12 (~ 45% mortality).”
Ln 212. It is more conventional to list p to 4 decimal places

Discussion
Ln 233. This is a great topic sentence if you split it in half. Start the next sentence “This affects their reproductive…” or similar.
Ln 236. Why not just say “For Decapods”
Ln 239. Why not just say “…environmental temperatures exert selective pressure on population phenotypes”
Ln 257. I suggest you not start a paragraph with “In turn”, because it suggests a link to a previous idea which is in a different paragraph. I would keep these terms for linking ideas directly. Why not just start the paragraph more simply with a good topic sentence, like “It has been well documented that contrasting temperatures…”
Ln 270. Why not just start “Temperature has also been shown to…”
Ln 274. Very interesting!
Ln 276. Good explanation
Ln 284. Why are you suddenly using the term molecules? Either remove or make consistent throughout the document

Conclusion
Ln 308. This is not bad but I would start the conclusions more formally, like “In our experiment where we did X and Y, differences were only observed in…”
Ln 312. Change to “These results suggest”
Ln 316. Change to “Overall,”
Ln 318. This point about the focal species as an ‘extremely important commercial resource’ should be made more of briefly in the manuscript before we get to the Conclusion section
Ln 323. I suggest you have another native English speaker go through the manuscript and improve readability. But, I will note that you did a good job here given English is not your first language.

Figures
Figures should be redrawn to be colorblind friendly (e.g. not red and blue)
Figure 3. I would apply a smoother to this plot but this is personal preference
Figures 5, 6. I would remove the grey gridlines from this plot with + theme_classic() but this is personal preference

Experimental design

The experimental design is appropriate

Validity of the findings

The findings are novel and will impactful for researchers interested in larval ontogeny of fished species

Additional comments

No additional comments

Reviewer 2 ·

Basic reporting

-

Experimental design

-

Validity of the findings

-

Additional comments

1. Tests should be conducted under at least 3 different criteria conditions.
2. Suggested test to determine the Influence of adding 1 criteria for temperature values between “26 - 30 °C”, so the tests were carried out at low to normal conditions in the wild.
3. In line 176 ‘Statistical analysis’ please add Regression Analysis to know the significance level between parameters on “Influence of temperature on developmental and biochemical traits of early ontogeny of the red squat lobster Grimothea monodon”

Reviewer 3 ·

Basic reporting

no comment

Experimental design

The experimental design could be more robust, with more temperature variations and also more Zoea V specimens so that the statistical data can be better analysed.

Validity of the findings

no comment

Additional comments

The manuscript is on Influence of temperature on developmental and biochemical traits of early ontogeny of the red squat lobster Grimothea monodon (H. Milne Edwards, 1837), and reports on the effect of temperature on Zoea I and V larval development (time, size and biomass), as well as CHN composition. Overall, the MS is well written and presented, but I feel that there are a number of criteria that should have been considered.
1 - Since there is an increase in temperature in the oceans and there is a hypothesis of tropicalisation of this environment, higher temperatures could have been proposed.
2 - Since the focus of the work is on the larval development of the species, it is important to know what happens to them. In addition, the number of Zoea V specimens is very low, so how can we ensure that we observe the effects at this stage?
3 - It would be interesting to carry out more robust multivariate analyses such as DistLM, RDA, GLM, canonical analysis to see how factors can contribute to larval development.
4 - The results of biomass, development, survival, and larval size need to be discussed.

Annotated reviews are not available for download in order to protect the identity of reviewers who chose to remain anonymous.

---

## Round 0.2 · accepted · Accept

All of the reviewers’ comments from the previous round have been satisfactorily addressed in your revised manuscript. As the previous reviewers were not available for this round, I have assessed the revision myself and find the current version to be clear, rigorous, and substantially improved. I am satisfied with the revisions and consider the manuscript ready for publication.